# Treatment outcomes among snakebite patients in north-west Ethiopia—A retrospective analysis

Inge Steegemans[1,2]*, Kassaye Sisay[3], Ernest Nshimiyimana[3], Gashew Gebrewold[4], Turid Piening[5], Endale Menberu Tessema[5], Birhanu Sahelie[5], Gabriel Alcoba[6], Fikre Seife Gebretsadik[7], Dirk Essink[2], Simon Collin[8], Emiliano Lucero[1], Koert Ritmeijer[1]

1 Médecins Sans Frontières, Amsterdam, The Netherlands, 2 VU University, Amsterdam, The Netherlands, 3 Médecins Sans Frontières, Abdurafi, Ethiopia, 4 Ethiopia Public Health Institute, Addis Ababa, Ethiopia, 5 Médecins Sans Frontières, Addis Ababa, Ethiopia, 6 Médecins Sans Frontières, Geneva, Switzerland, 7 Federal Ministry of Health, Addis Ababa, Ethiopia, 8 Federal University of Espírito Santo, Vitória, Brazil

* ingesteegemans@gmail.com

**Data Availability Statement:** MSF has a managed access system for data sharing that respects MSF's legal and ethical obligations to its patients to collect, manage and protect their data

## Abstract

### Background

Millions of people are bitten by venomous snakes annually, causing high mortality and disability, but the true burden of this neglected health issue remains unknown. Since 2015, Médecins Sans Frontières has been treating snakebite patients in a field hospital in northwest Ethiopia. Due to the poor market situation for effective and safe antivenoms for Sub-Saharan Africa, preferred antivenom was not always available, forcing changes in choice of antivenom used. This study describes treatment outcomes and the effectiveness and safety of different antivenoms used.

### Methodology / Principal findings

This retrospective observational study included 781 snakebite patients presenting at the field hospital between 2015 and 2019. Adjusted odds ratios, 95%-CI and p-values were used to compare the treatment outcome of patients treated with Fav-Afrique (n = 149), VacSera (n = 164), and EchiTAb-PLUS-ICP (n = 156) antivenom, and to identify the risk of adverse reactions for each antivenom. Whereas only incidental snakebite cases presented before 2015, after treatment was made available, cases rapidly increased to 1,431 in 2019. Envenomation was mainly attributed to North East African saw-scaled viper (*Echis pyramidum*) and puff adder (*Bitis arietans*). Patients treated with VacSera antivenom showed lower chance of uncomplicated treatment outcome (74.4%) compared to Fav-Afrique (93.2%) and EchiTAb-PLUS-ICP (90.4%). VacSera and EchiTAb-PLUS-ICP were associated with 16- and 6-fold adjusted odds of treatment reaction compared to Fav-Afrique, respectively, and VacSera was weakly associated with higher odds of death.

### Conclusions / Significance

Snakebite frequency is grossly underreported unless treatment options are available. Although EchiTAb-PLUS-ICP showed favorable outcomes in this retrospective analysis,

responsibility. Ethical risks include, but are not limited to, the nature of MSF operations and target populations being such that data collected are often highly sensitive. Data are available on request in accordance with MSF's data sharing policy (available at: http://fieldresearch.msf.org/msf/handle/10144/306501). Requests for access to data should be made to data.sharing@msf.org.

**Funding:** The author(s) received no specific funding for this work. There were no extra costs involved.

**Competing interests:** The authors have declared that no competing interests exist.

prospective randomized trials are needed to evaluate the effectiveness and safety of the most promising antivenoms for Sub-Saharan Africa. Structural investment in sustained production and supply of antivenom is urgently needed.

## Author summary

Snakebite is an important but highly neglected medical problem, causing high morbidity and mortality in low-and-middle income countries all over the world, including Ethiopia. When bitten by a venomous snake, appropriate treatment with the correct antivenom is crucial. However, in Ethiopia the correct antivenom is not always available due to low access to health facilities in rural areas. Even if antivenom is available, patients can often not afford it due to the high costs of clinical effective antivenoms. This study explores the effectiveness of three different antivenoms, Fav-Afrique, VacSera and EchiTAb-PLUS-ICP, and provides insights in factors that might influence the outcome of the snakebite treatment. This study shows that patients treated with VacSera antivenom have a higher chance of an adverse outcome of the treatment. Much more research on snakebite epidemiology and the effectiveness of antivenoms should be conducted, aiming to alert governments and antivenom manufacturers that this is a serious problem which cannot be ignored any longer.

## Introduction

Each year, around 5.5 million people worldwide are bitten by snakes, causing approximately 100,000 deaths and leaving 400,000 people disabled [1,2]. In Sub-Saharan Africa, snakebites are causing an estimated 435,000–580,000 envenomations and 20,000–32,000 deaths annually [3]. Snakebites mostly affect the poorest populations [4,5]. People living in rural areas are most susceptible as they live and work in areas where snakes reside [6]. In combination with poor access to health services, scarcity of safe and effective antivenoms and the economic impact of disabled young victims, snakebites put a high burden on the population and the economy of low-and-middle-income countries [7].

Data regarding the incidence and mortality of snakebites is often inaccurate due to the under-reporting of snakebites by patients and health institutions [2,8,9]. Snakebite victims often do not seek medical care in government dispensaries but go to traditional healers instead [2,9]. The reasons for this are low access to health facilities in rural areas and medical treatment being too expensive [2,10,11]. Moreover, antivenom treatment is often not available [12]. Antivenoms are the only effective treatment against envenomation when someone is bitten by a venomous snake [4]. They can prevent or reverse most toxic effects that occur during snakebite envenoming and they play a significant role in limiting the morbidity and mortality. However, at this moment antivenoms are mainly equine immunoglobulins, which are snake-specific and can cause serious adverse events; when an unsuitable product is used which has not been developed to neutralize venoms of relevant endemic snake species, it is ineffective [13,14]. Due to the high costs of clinical effective antivenoms, many cheaper but low-quality or even counterfeit antivenoms are circulating on the pharmaceutical market, posing a threat to populations of many low-and-middle-income countries [15]. The high costs of antivenoms also lead to a low demand, which makes it a non-profitable market for manufacturers which has resulted in a shortage in the supply of antivenoms [12].

In order to develop strategies for the management of snake antivenoms, epidemiological data is essential [1,8]. However, there is still a lack of reliable data concerning the incidence, burden and the management of snakebites in different regions of the world [1,2,8]. Moreover, limited data is available on the effectiveness and safety of different antivenoms in Sub-Saharan Africa [3]. With this information at hand, manufacturers of antivenoms will be able to better manage the production, and health authorities will be able to distribute the most effective antivenoms to the right places [12]. Recent literature states that population-based studies researching the incidence, morbidity, and mortality in high burden countries are required urgently [8].

Ethiopia is one of many countries facing a high incidence of snakebites and low access to treatment [16]. In the north-western lowlands of Ethiopia, snakebites occur regularly and present a sizeable health hazard. Since 2015, Médecins Sans Frontières (MSF) has been treating snakebite victims with antivenom in a field hospital in Abdurafi, in the lowlands of the Amhara region in north-west Ethiopia. The large commercial farming schemes in this region attract over 350,000 seasonal migrant workers from other parts of Ethiopia and neighboring countries [17,18]. Due to the poor conditions on these farms, migrant workers are particularly vulnerable to snakebites as they work, live, and sleep in the bare fields without protection.

MSF has been providing antivenom treatment for snakebite patients since March 2015. Before antivenom was available, standard care for snakebite patients consisted of first aid, supportive care and management, and mental support. In case a patient was in need of more intensive treatment, such as surgery, the clinic provided referral to a secondary or tertiary hospital. Since March 2015, four different antivenoms were used by MSF to treat snakebite envenoming: Fav-Afrique (March '15—Aug '16), VacSera (Jan '17 –Aug '18), and EchiTAb-PLUS-ICP and SAIMR (since Sep '18) (S1 Table). The different antivenoms were used based on accessibility and availability. First, the Fav-Afrique antivenom (Sanofi Pasteur, France) was used, which is a polyvalent antivenom and is effective against most bites of medically important snakes in sub-Saharan Africa *(Bitis spp.*, *Dendroaspis spp.*, *Echis spp.*, *Naja spp.)* [19]. Because the production of Fav-Afrique was stopped in 2014, and the last batch expired in June 2016, another antivenom was urgently needed. At that time, the Egyptian VacSera antivenom (VACSERA, Egypt) was the only antivenom licensed for importation in Ethiopia. VacSera antivenom is also a polyvalent antivenom and is raised against endemic snakes in Egypt such as spitting cobras (*Naja haje*), horned vipers (*Cerastes cerastes*), saw-scaled vipers (*Echis coloratus*), and puff adders (*Bitis arietans*) [20]. However, it is not raised against the East African carpet viper (*Echis pyramidum*), which seems to be the most frequent causative species in the lowlands of north-west Ethiopia. The third antivenom protocol which is still being used includes EchiTAb-PLUS-ICP (ICP, Costa Rica) and SAIMR polyvalent antivenom (SAVP, South Africa). EchiTAb-PLUS-ICP antivenom is a polyvalent antivenom and is used in case of envenoming by puff adders (*B. arietans*), West African carpet vipers (*Echis ocellatus*), and spitting cobras (*Naja nigricolis*). SAIMR is an antivenom used in case of neurotoxic envenoming by mambas (*Dendroaspis spp.*), spitting and non-spitting cobras (*Naja spp.*) and in critical cases of severe cytotoxic envenomation by *Bitis* vipers.

Whether treatment of snakebite envenoming is effective depends highly on the early administration of adequate amounts of the correct antivenom to neutralize venom antigens and reverse its effects [21–23]. However, in Ethiopia, antivenom is scarcely available in public hospitals, and if it is, it is expensive due to market failure and little attention to snakebite envenoming [2,9]. This study provides epidemiological data on the frequency and management of snakebites in north-west Ethiopia and aims to contribute to the current knowledge about the treatment of snakebites by describing which antivenom is most effective and safe, and what factors influence the treatment outcome. This context-specific information aims to contribute to the improvement of snakebite patient care and its medical management in Ethiopia.

## Methods

### Ethics statement

This research fulfills the exemption criteria set by the Médecins Sans Frontières Ethics Review Board (MSF ERB) for a posteriori analysis of routinely collected clinical data and therefore does not require MSF ERB review. The research was conducted with permission from MSF Holland. In addition, this research and the method of data collection has been approved by the Ethiopian Public Health Institute–Institutional Review Board (EPHI—IRB).

### Design and research setting

In this descriptive retrospective cross-sectional study, secondary data analysis was performed with data from the MSF field hospital in Abdurafi, West Armachiho district, Amhara region in Ethiopia. In this hospital, snakebite patients are admitted for observation and treatment. A snakebite management algorithm is used to determine the treatment a patient requires. This algorithm is based on a syndromic approach as in most cases reliable snake information such as the species is unknown. There are three different types of snakebite syndromes: cytotoxic, haematotoxic, and neurotoxic. The cytotoxic syndrome affects skin and muscle cells, leading to extensive swelling, blistering, muscle damage, and possibly local tissue necrosis. The haematotoxic syndrome affects the cardiovascular system and blood functions and often causes internal and external bleeding. The neurotoxic syndrome affects the nervous system and causes flaccid muscular paralysis [24,25]. The algorithm indicates if antivenom should be administered or not and what kind of supportive care is recommended, based on the syndrome, its severity, and a blood clotting test. Antivenom is not administered when symptoms are mild, e.g. non-progressive swelling in the extremities or local bleeding, and when there are no critical signs.

Suspected snake species was assessed by patient description of the snake if possible, using a snake atlas with pictures and sizes. Snakes caught by a snake catcher of the Ethiopian Public Health Institute on the farms in Abdurafi area in 2017 identified *Echis pyramidum* and *Bitis arietans* as the main venomous snakes in the area.

### Sampling procedure

The study population consists of patients who have attended the Abdurafi field hospital due to a snakebite incident between January 2015 and December 2019. Patients were divided into those who received antivenom and those who did not receive antivenom. Every patient who received antivenom between 2015 and 2019 was included in the study.

For the patient group who did not receive antivenom treatment, Grays' systematic sampling strategy was used where every third chronological patient was included in the study [26]. Patients who were admitted to the clinic during the period from September 1st 2016 until December 31th 2016 were excluded from the analysis in order to exclude possible bias, since there was no antivenom available during this period. Patients who received antivenom were categorized based on the different antivenoms that were used during the research period: Fav-Afrique, VacSera, and EchiTAb-PLUS-ICP. Since there was only one patient during the research period who was treated with SAIMR antivenom, SAIMR antivenom was excluded from the analysis.

### Treatment outcome and adverse event parameters

A patient was considered to have an adverse treatment outcome when the patient died, or has permanent physical damage or debilitation, for example, the amputation of a limb, or organ

failure, or if the patient experienced an adverse treatment reaction, defined as an acute (anaphylactic or pyrogenic) reaction to antivenom presenting symptoms such as rash, itching, nausea, headache, fever, wheezing, body aches, hypotension, cyanosis, and altered level of consciousness, which develop up to one hour during or after the administration of antivenom [27]. All other patients were considered to have an uncomplicated outcome. Patients were classified into the two outcome groups using routinely collected data as described below.

## Data collection and analysis

Routinely collected data were extracted from MSF standard snakebite patient forms (S1 Form) and corresponding patient files kept at the Abdurafi field hospital. Patients admitted to the field hospital were monitored until they were discharged. Anonymized data were digitalized and analyzed using IBM SPSS Statistics 23 and Stata 15 (StataCorp. 2017. Stata Statistical Software: Release 15. College Station, TX: StataCorp LLC). Epidemiological characteristics and outcomes were investigated using the following variables: age, gender, time between snakebite and admission to hospital, bite site, location, context of the bite, snake species, type of envenomation, number of antivenom vials received, time between admission and administration of antivenom, time between antivenom administration and restored blood coagulation, adverse reactions to antivenom, signs of severe envenomation, duration of hospitalization and whether the patient survived. Continuous variables were summarized using the median and interquartile range (IQR). Categorical and dichotomous variables are summarized using the number of patients (n) and percentages. Chi-squared tests (or Fisher's exact tests if any frequency was <5) for categorical variables and Kruskal-Wallis tests for continuous variables were used to compare the differences between groups (alpha = 0.05). Odds ratios for adverse compared with uncomplicated outcomes among patients receiving different types of antivenom were estimated using logistic regression, adjusted for age, sex and time from bite to admission (as *a priori* confounders) and other variables with complete data and associated with antivenom type and adverse outcome (alpha = 0.10) and not considered to be on a causal pathway.

## Results

The frequency of reported snakebites in the Abdurafi field hospital from 2015 to 2019 shows a seasonal pattern (Fig 1), peaking between June and November, which coincides with the agricultural season and presence of migrant workers. The number of admissions during 2016 was less than during 2015 due to a long rupture period of supply of antivenom in the hospital during the 2016 peak season after the expiry of Fav-Afrique. The annual drop in cases presenting in September is associated with the Ethiopian New Year (*Enkutatash*) when many migrant workers go back to their families for the celebrations. Overall, there is a clear increasing trend over the years in the total number of snakebite patients admitting to the field hospital. In 2015 this total was 330, in 2016 the total dropped to 159 as mentioned. In 2017 322 patients admitted to the clinic, in 2018 this doubled to 647 patients and in 2019 there was a total of 1,431 patients admitted to the clinic. During the period that Fav-Afrique was available (2015–2016) 32.9% of all snakebite admissions were treated with antivenom. This reduced to 27.6% during the VacSera period (2017–2018), and 20.5% during the EchiTAb-PLUS-ICP period (2018–2019).

Out of the 2,889 snakebite patients admitted in Abdurafi between 2015 and 2019, 781 patients were included in the study of which 149 in the Fav-Afrique group, 164 in the VacSera group, 156 in the EchiTAb-PLUS-ICP group, and 312 in the no-antivenom group. The characteristics of the patients and bites are presented in Table 1. The median ages of the different

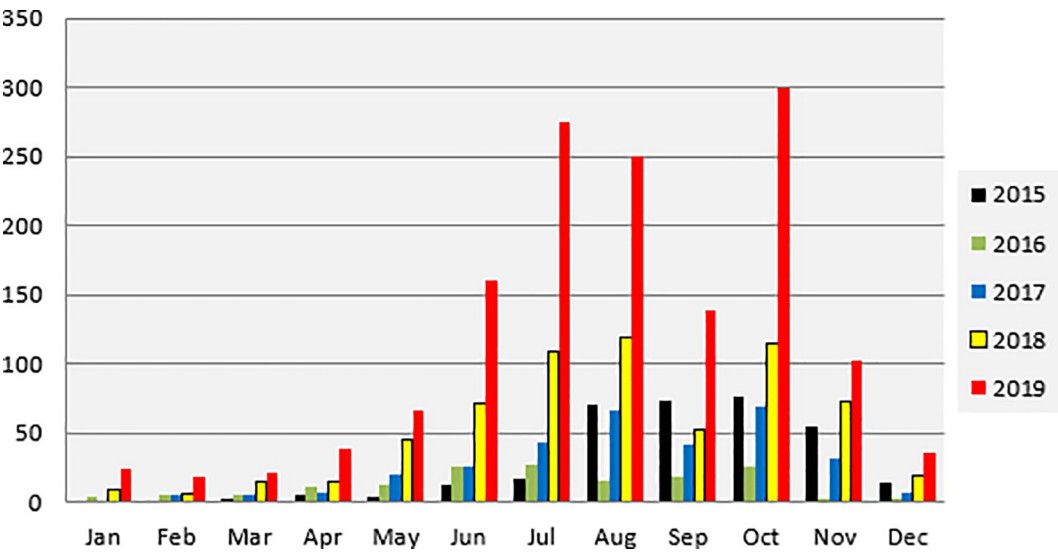

**Fig 1. Monthly snakebite admission to Abdurafi field hospital, 2015–2019.**

groups were between 24–26 years old (range 1–74 years) and 85 to 90 percent of the patients in the different antivenom groups were male. The median time between bite and admission was 10 hours, 7.4 hours, 7.7 hours, and 7 hours for the Fav-Afrique group, VacSera group, Echi-TAb-PLUS-ICP group, and no-antivenom group respectively. In each group, most of the patients were bitten in the foot or leg (54–79%) and most bites occurred on the field (44–65%) while working (37–60%). Within each group, most victims did not know which snake had caused the bite, but most patients showed both cytotoxic and haematotoxic symptoms (57–99%). Patients in the no-antivenom group showed symptoms of envenoming as well, but only in mild forms: only local swelling and no signs of active bleeding. As shown in Table 2, the median time between admission to the clinic and the administration of antivenom for Fav-Afrique was 1.5 hours. For VacSera this time was 6 hours and for EchiTAb-PLUS-ICP this was almost 9 hours. The highest rate of adverse antivenom reactions occurred within the group of patients treated with VacSera (18.3% vs. 1.3% and 7.1% for patients treated with Fav-Afrique and EchiTAb-PLUS-ICP respectively). Also, patients treated with VacSera developed more signs of severe envenomation (22.0%) compared to Fav-Afrique (16.1%) and EchiTAb-PLU-S-ICP (2.6%). In total, 63 patients experienced intense pain (requiring management with analgesics stronger than paracetamol), 61 patients experienced extensive internal or external bleeding (bleeding not only from the bite site but also from other sites of the body), 46 patients experienced intense swelling (swelling extending beyond one joint) and 29 patients experienced fever/infection. These were the main signs of severe envenomation during the treatment. There were no amputations recorded in the database in any treatment group. In this population, uncomplicated outcomes were observed less frequently for VacSera (74.4%) than for Fav-Afrique (93.2%) and EchiTAb-PLUS-ICP (90.4%). The mortality rate was highest in the group treated with VacSera (4.9%) compared to 1.9% and 1.3% in the EchiTAb-PLUS-ICP group and the Fav-Afrique group. In patients who did not receive antivenom 1.6% experienced an adverse outcome, but no deaths.

Among patients who received antivenom, those treated with VacSera or EchiTAb-PLU-S-ICP had almost 6-fold and 3-fold odds respectively of experiencing an adverse outcome compared with those who received Fav-Afrique after adjusting for age, sex, signs of severe

**Table 1. Characteristics of snakebite patients admitted to Abdurafi field hospital between 2015 and 2019.**

| | Fav-Afrique (n = 149) | VacSera (n = 164) | EchiTAb-PLUS-ICP (n = 156) | No antivenom (n = 312) |
|---|---|---|---|---|
| **Age (years)** | 24 (20–30) | 24 (20–30) | 26 (20–35) | 24 (20–30) |
| **Gender (m) n (%)** | 137 (91.9) | 147 (89.6) | 132 (84.6) | 275 (88.1) |
| **Bite site n (%)** | | | | |
| Foot | 65 (43.6) | 109 (66.5) | 119 (76.3) | 158 (50.6) |
| Leg | 16 (10.7) | 10 (6.1) | 4 (2.6) | 29 (9.3) |
| Hand | 54 (36.2) | 42 (25.6) | 27 (17.3) | 103 (33.0) |
| Arm | 7 (4.7) | 0 (0.0) | 0 (0.0) | 9 (2.9) |
| Other | 7 (4.7) | 3 (1.8) | 6 (3.8) | 13 (4.2) |
| **Location of bite n (%)** | | | | |
| Field | 66 (44.3) | 97 (59.1) | 101 (64.7) | 180 (57.7) |
| Compound | 4 (2.7) | 14 (8.5) | 16 (10.3) | 19 (6.1) |
| Indoors | 3 (2.0) | 12 (7.3) | 3 (1.9) | 18 (5.8) |
| Road | 3 (2.0) | 8 (4.9) | 27 (17.3) | 8 (2.6) |
| Missing | 73 (49.0) | 33 (20.1) | 9 (5.8) | 87 (27.9) |
| **Context of bite n (%)** | | | | |
| Working | 55 (36.9) | 98 (59.8) | 74 (47.4) | 168 (53.8) |
| Walking | 7 (4.7) | 37 (22.6) | 47 (30.1) | 46 (14.7) |
| Sleeping | 12 (8.1) | 11 (6.7) | 21 (13.5) | 26 (8.3) |
| Playing | 3 (2.0) | 12 (7.3) | 8 (5.1) | 8 (2.6) |
| Other | 1 (0.7) | 1 (0.6) | 2 (1.3) | 2 (0.6) |
| Missing | 71 (47.7) | 5 (3.0) | 4 (2.6) | 62 (19.9) |
| **Suspected snake species n (%)** | | | | |
| *Echis* | 5 (3.4) | 83 (50.6) | 53 (34.0) | 105 (33.7) |
| *Bitis* | 0 (0.0) | 2 (1.2) | 34 (21.8) | 3 (1.0) |
| *Dendroaspis* | 1 (0.7) | 8 (4.9) | 0 (0.0) | 6 (1.9) |
| Other | 0 (0.0) | 3 (1.8) | 7 (4.5) | 28 (9.0) |
| Unknown | 26 (17.4) | 60 (36.6) | 60 (38.5) | 102 (32.7) |
| Missing | 117 (78.5) | 8 (4.9) | 2 (1.3) | 68 (21.8) |
| **Type of envenomation n (%)** | | | | |
| Cytotoxic | 2 (1.3) | 17 (10.4) | 51 (32.7) | 113 (36.2) |
| Haematotoxic | 0 (0.0) | 14 (8.5) | 15 (9.6) | 6 (1.9) |
| Cytotoxic + haematotoxic | 142 (95.3) | 132 (80.5) | 80 (51.3) | 192 (61.5) |
| Cytotoxic+ neurotoxic | 0 (0.0) | 0 (0.0) | 5 (3.2) | 0 (0.0) |
| Cytotoxic + haematotoxic + neurotoxic | 5 (3.4) | 1 (0.6) | 5 (3.2) | 1 (0.3) |

envenomation, body part bitten, heart rate and temperature on admission, and time from bite until admission (Table 3).

The negative confounding of the effects of the different antivenoms observed in the multivariable model was cumulative and not attributable to any particular covariate. Of these, only severe envenomation was associated with odds of adverse outcome (odds ratio = 8.84, 95% CI 4.38–17.8). Complete data were available for all N = 466 patients. The model did not adjust for time from admission to administration of antivenom because these data were missing for 167/468 patients. Odds ratios for treatment reactions and death by antivenom type are shown in Table 4. VacSera and EchiTAb-PLUS-ICP were associated with 16- and 6-fold odds of treatment reaction, respectively, and VacSera was weakly associated with higher odds of death. However, all estimates were imprecise (very wide confidence intervals) because of the small numbers of patients experiencing these specific outcomes.

**Table 2. Treatment and outcomes of snakebite patients admitted to Abdurafi field hospital between 2015 and 2019.**

| | Fav-Afrique (n = 149) | VacSera (n = 164) | EchiTAb-PLUS-ICP (n = 156) | No antivenom (n = 312) | P-value* |
|---|---|---|---|---|---|
| **Time from bite until admission in hours (IQR)** | 10.0 (3.9–21.0) | 7.4 (3.8–21.0) | 7.7 (3.8–18.7) | 7.0 (4.0–17.0) | 0.519 |
| **Time from admission to AV administration in hours (IQR)** | 1.5 (0.5–11.3), n = 54 | 6.0 (0.3–23.3), n = 131 | 8.8 (0.0–24.4), n = 136 | - | 0.319 |
| **Total time from bite to AV administration in hours (IQR)** | 18.4 (10.3–37.9), n = 54 | 25.0 (13.0–43.0), n = 131 | 22.2 (12.5–48.9), n = 136 | - | 0.284 |
| **Number vials of AV administered** | 2 (2–6)** | 4 (4–22)*** | 3 (3–9)** | - | <0.001 |
| **Time until blood clotting restoration in days (IQR)** | 2 (1–3) | 2 (1–3) | 1 (0–2) | 1 (1–2) | 0.066 |
| **Severe envenomation n (%)** | 24 (16.1) | 36 (22.0) | 4 (2.6) | 17 (5.4) | <0.001 |
| **Hospital duration in days (IQR)** | 4 (3–5) | 4 (3–6) | 4 (3–6) | 2 (1–3) | <0.001 |
| **Outcome n (%)** | | | | | |
| Uncomplicated | 138 (93.2) | 122 (74.4) | 141 (90.4) | 299 (98.4) | |
| Antivenom reaction | 2 (1.3) | 30 (18.3) | 11 (7.1) | - | |
| Referred | 6 (4.0) | 9 (5.5) | 1 (0.6) | 5 (1.6) | |
| Died | 2 (1.3) | 8 (4.9) | 3 (1.9) | 0 (0.0) | <0.001 |
| Adverse outcome | 10 (6.8) | 42 (25.6) | 15 (9.6) | 5 (1.6) | <0.001 |
| Discharge against advice | 1 (0.7) | 0 (0.0) | 0 (0.0) | 8 (2.6) | |

* Chi-squared tests (or Fisher's exact tests if any frequency was <5) for categorical variables and Kruskal-Wallis tests for continuous variables. Data complete for each variable in each group unless otherwise indicated

**Presenting the range instead of IQR, missing for 100/156 patients who received EchiTAb-PLUS-ICP.

Abbreviations: AV = Antivenom; IQR = Inter Quartile Range.

## Discussion

Over the past five years, much-needed information was gathered about the epidemiology of snakebites and the use of antivenoms in north-west Ethiopia. The results of this study show that over the years, there is an exponential growth of snakebite patients presenting at the MSF hospital since antivenom treatment was made available. This is a result of spreading awareness of treatment availability in Abdurafi among the migrant and resident population in the region, which led to an expanding catchment area. The improvement of communication networks, such as cell phone and internet services, and infrastructure (roads and vehicle availability) is also contributing to increased health-seeking behavior and improved access to Abdurafi MSF field hospital. But it also indicates massive underreporting of snakebites when comprehensive snakebite management is not available [28].

Monthly admissions to the hospital between 2015 and 2019 show a clear seasonal trend, with most patients admitted between June and November. These are the most labor-intensive

**Table 3. Odds of adverse outcome in snakebite patients treated with different antivenoms at Abdurafi field hospital between 2015 and 2019.**

| Antivenom | n | Adverse outcome (%) | Odds ratio (95% CI) | p-value | Adjusted odds ratio*** (95% CI) | p-value |
|---|---|---|---|---|---|---|
| Fav-Afrique | 148* | 10 (6.8%) | 1.00 (reference) | | | |
| VacSera | 164 | 42 (25.6%) | 4.72 (2.27–9.80) | <0.001 | 6.16 (2.64–14.4) | <0.001 |
| EchiTAb-PLUS-ICP | 155** | 15 (9.6%) | 1.47 (0.64–3.38) | 0.367 | 3.16 (1.12–8.00) | 0.029 |

* excluding one patient who defaulted treatment (discharged against medical advice)

** excluding one patient who had missing data for heart rate

*** adjusted for age, sex, and time from bite until admission (*a priori*) and signs of severe envenomation, body part bitten, heart rate and temperature on admission (associated with outcome and antivenom)

**Table 4. Odds of treatment reaction and death in snakebite patients treated with different antivenoms at Abdurafi field hospital between 2015 and 2019.**

| Antivenom | n | Outcome (%) | Odds ratio (95% CI) | p-value | Adjusted odds ratio** (95% CI) | p-value |
|---|---|---|---|---|---|---|
| | | Treatment reaction | | | | |
| Fav-Afrique | 148* | 2 (1.4%) | 1.00 (reference) | | | |
| VacSera | 164 | 30 (18.3%) | 16.3 (3.83–69.7) | <0.001 | 16.2 (3.79–69.2) | <0.001 |
| EchiTAb-PLUS-ICP | 156 | 11 (7.1%) | 5.54 (1.21–25.4) | 0.028 | 5.78 (1.25–26.7) | 0.024 |
| | | Death | | | | |
| Fav-Afrique | 148* | 2 (1.4%) | 1.00 (reference) | | | |
| VacSera | 164 | 8 (4.9%) | 3.77 (0.79–18.0) | 0.097 | 4.56 (0.90–23.0) | 0.066 |
| EchiTAb-PLUS-ICP | 156 | 3 (1.9%) | 1.44 (0.24–8.75) | 0.691 | 1.16 (0.18–7.34) | 0.871 |

* excluding one patient who defaulted treatment (discharged against medical advice)

** adjusted for age, sex, and time from bite until admission

months of the agricultural season when workers are in the fields for planting, weeding, and harvesting [22]. Protective means, like boots and gloves, are not made available, and would most likely have poor acceptability because of inconvenience due to the hot climate, with temperatures above 30˚C. The significant relative decrease in patient numbers presenting in the period from July 2016 to January 2017 is associated with the expiry of Fav-Afrique antivenom in September 2016 and no suitable alternative antivenom being available until January 2017. This event demonstrated the quickly spreading awareness among the population that there was no antivenom available to treat snakebites anymore, leading to a change in the health-seeking behavior.

Most patients admitted to the hospital showed both cytotoxic and haematotoxic symptoms, which could indicate certain snake species being responsible for the bite, notably the Egyptian saw-scaled viper or Northeast African carpet viper (*E. pyramidum*) and puff adder (*B. arietans*). There were many patients who admitted to the hospital with both cytotoxic and haematotoxic symptoms who did not receive antivenom. In most of these cases, the envenomation was mild and no antivenom was required. Few patients admitted to the hospital with neurotoxic symptoms. However, the low number of patients presenting with neurotoxic symptoms does not necessarily indicate a low incidence of neurotoxic bites in the region. Because neurotoxic envenoming by elapids can cause respiratory paralysis and death within hours, patients may not have been able to access healthcare in time [29]. The presence of elapids (Egyptian cobra (*N. haje*) and black mamba (*dendroaspis polylepis*)) has been reported in the north-western Ethiopian lowlands [30]. In Table 1 there are six patients in the group who did not receive antivenom, who stated that they were bitten by a mamba. This is remarkable as these bites are typically very severe neurotoxic bites. An explanation might be that these were cases of misidentification of the snake species. This happens frequently, especially since many patients are bitten at night, which is also one of the reasons why it is difficult to identify the species of snakes in the area causing snakebites. To address this problem, a snake atlas was introduced in 2017 including pictures and descriptions of endemic venomous and non-venomous snakes to help patients identify the snake that bit them.

The results of this study are not representative for the whole of Ethiopia. Ethiopia has a wide topographical diversity, ranging from desert, savannah, grassland and tropical forests to alpine mountains. Therefore, the snake epidemiology in Ethiopia varies greatly between different regions [31]. However, the results are representative for the lowland savannah region in north-west Ethiopia.

The main signs of severe envenomation that occurred during the treatment were severe pain, extensive internal or external bleeding, severe swelling, and fever/infection of the bite

site. However, in most cases, these severe manifestations can be attributed to the venom-induced syndrome. More severe complications such as massive hemorrhage, coma, respiratory distress, necrosis and organ failure were most frequent in the patient group which was treated with Vac-Sera. Patients whose condition was very critical were referred to a tertiary hospital in Gondar.

We compared the occurrence of adverse outcomes according to the three different antivenoms used; Fav-Afrique, VacSera, and EchiTAb-PLUS-ICP. Fav-Afrique was found to be a safe and effective antivenom, which is in line with other authors' findings [19], and the number of adverse reactions was lower than observed for VacSera and EchiTAb-PLUS-ICP.

VacSera had the highest rate of adverse reactions. This outcome corresponds with previous research by Larnyang, Salako & Alder et al., who found that VacSera caused adverse reactions in five out of six patients, and stated that VacSera did not fulfill the efficacy and safety criteria [28,32]. EchiTAb-PLUS-ICP showed to be safe due to the low rates of adverse reactions. This is in line with neutralization studies which have also demonstrated the safety and efficacy of EchiTAb-PLUS-ICP in African viperid envenoming [28,33]. However, more clinical trials evaluating the safety and effectiveness of the EchiTAb-PLUS-ICP antivenom need to be conducted to confirm these results [34,35].

Due to the low number of deaths it was not possible to compare the mortality rates between the antivenoms. There was no mortality in the group of patients who did not receive antivenom. This is explained by the fact that patients in the no-antivenom group by definition suffered from dry bites or mild envenoming. Patients treated with antivenom were more severely envenomed and therefore in need of antivenom. Out of 13 patients who died, 8 died within 48 hours after admission, indicating that these patients presented with severe envenoming syndrome on admission. Frequent signs and complications in the patients who died were organ failure, coma, hemorrhage and respiratory distress. Two patients who died experienced a shock after administration of VacSera antivenom.

The average time between bite and hospital admission decreased from 10 hours during the time that Fav-Afrique was available (2015–2016) to 7–8 hours in the later periods. This may reflect increased awareness among the communities and earlier presentation. However, in this study the time between the bite and admission to the hospital was not associated with odds of an adverse outcome of the treatment. This finding is contrary to previous studies which showed that delay in treatment is strongly associated with an adverse outcome of the treatment and that mortality is higher among patients who present late compared to patients who present within 24 hours after the bite [21,36–40].

Interestingly, the median time between admission and antivenom administration increased over time from 1.5 hours in 2015–2016 to 8.8 hours during 2018–2019. However, in this study there was no association between the time between admission and the administration of antivenom and the development of an adverse outcome of the treatment. This could be explained by the improved clinical monitoring of envenoming effects and management of severe envenomation before administering antivenom. This may have resulted in reduced indications for antivenom treatment, without negatively impacting treatment outcome. This is reflected in the proportion of patients admitted that received treatment with antivenom, which decreased significantly over time, from 32.9% during the Fav-Afrique period (2015–2016) to 27.6% during the VacSera period (2017–2018) and 20.5% during the EchiTAb-PLUS-ICP period (2018–2019). Although there were no major changes in first aid care over the years, over time patients did present earlier due to community awareness. This allowed the healthcare workers at the clinic to take more time for patient observation and to decide whether the patient needed antivenom or not.

Other characteristics, such as bite site, the context of the bite, snake species responsible, and type of envenomation, were not association with adverse outcomes of the treatment. The fact

that snake species had no effect could be explained by the fact that the antivenoms used are effective against the toxins of the specific snakes living in this certain geographical area. Snake species that are causing the greatest burden in this area are the Egyptian saw-scaled viper (*E. pyramidum*) and puff adder (*B. arietans*) [2]. The outcome of the treatment and the effectiveness of the antivenom can vary between different geographical areas as the venom composition can differ greatly within one certain snake species. These variations can introduce major medical implications to the treatment of snakebite victims. Also, depending on the snake species, different symptoms of envenoming can appear [41,42]. This is also the reason why it is important to know which species caused the envenoming since the antivenom might not be raised against the venom of certain snake species from that region. However, in all three groups of antivenom, there was a large proportion of patients who were not able to identify the type of snake species from the atlas.

Despite the production of Fav-Afrique antivenom was ceased by Sanofi-Pasteur in 2014, it remains a relevant product since MicroPharm (UK) announced its intention to relaunch the product after Sanofi-Pasteur signed an agreement to divest its antivenom immunoglobulin range to MicroPharm in December 2017 [43] Micropharm and Sanofi-Pasteur are currently working on the technology transfer.

Although antivenom affordability is a major factor in patient access to appropriate snakebite treatment, affordability was not an issue in this study. All antivenom treatment and supportive care and hospitalization were provided free of charge at the MSF clinic.

This is one of few studies on epidemiology of snakebites and treatment outcomes of various antivenoms in clinical practice in Ethiopia using a relatively large patient cohort. However, as this study is not a randomized controlled trial evaluating different antivenoms, it is important to acknowledge limitations. Firstly, since this is mainly a descriptive study, any comparisons between the antivenoms are indications since the antivenoms were used in different periods, introducing possible bias to the comparisons by differences in patient population or snakebite management practices over time. Although we adjusted for *a priori* and actual confounders, our estimates will still be subject to residual and unmeasured confounding. Similarly, although data were mostly complete for all variables in our final model, our analytical dataset represented a modest proportion of all patients treated during the study period.

Secondly, since this is a retrospective study with routinely collected data, there may have been factors that could have a possible influence on the outcome of the treatment which were not captured by the routine data collection. For example, in the data, there is no information about the consultation of traditional healers or the use of traditional remedies by the patients. Common traditional healing practices may include cutting, suction, tourniquets, black stones, local incisions, and herbal-based medicines which are put inside the wound or are taken orally by the patient [2]. For further research, it is recommended to include this in the data collection. While the use of traditional first aid may not directly predict an adverse outcome or mortality, some of these practices are harmful, and in addition cause a delay in seeking treatment, which could result in an increased occurrence of complications [2,4,21,36,37].

With the results of this research, a piece of the burden of snakebites in Ethiopia is revealed which shows that snakebites are a huge public health problem that deserves more attention. It gives an overview of the size of the problem in this part of Ethiopia and provides information necessary for improving access to antivenom treatment, and training of health staff on appropriate first aid and management of signs of severe envenoming and adverse reactions. Antivenom protocols can be adapted based on the information provided and snakebite management can be improved. Also, health promotion and advocacy among the communities at risk is essential to prevent snakebites, and to improve health-seeking behavior in the case of snakebites. However, in order to address the large public health problem caused by snake

envenoming across Sub-Saharan Africa, further research is needed to determine the efficacy, safety, and regional specificity of existing antivenoms for Sub-Saharan Africa, linked with structural investments in sustained production and supply of antivenom.

## Supporting information

**S1 Table. Table of antivenoms.**
(PDF)

**S2 Table. Full analyses Table 3.**
(PDF)

**S3 Table. Full analyses Table 4.**
(PDF)

**S1 Form. Snakebite Patient Form.**
(PDF)

## Acknowledgments

We gratefully acknowledge the assistance of the Ethiopia Public Health Institute and the Federal Ministry of Health of Ethiopia. In addition, we wish to acknowledge the following individuals.

We would like to thank the nurses: Mekuanint Anagaw, Kasanesh Cherie, Umer Ibrahim, Woldegebriel, Teferi Molla, Taddelu Ajamu, Muluken Alemu, Molla Dessie, Fikremariam Yibabie, Tewodrose Dagnew and Eshetie Gobie, and the clinical officers: Leya Haileyesus and Seble Bihonegen who were working at the ward treating snakebite patients. They were able to provide us with valuable inside information on snakebite treatment and shared patient experiences. The PI would like to thank Gijs Hovenkamp for his continuous support during the writing of this manuscript.

## Author Contributions

**Conceptualization:** Inge Steegemans, Dirk Essink, Emiliano Lucero, Koert Ritmeijer.

**Data curation:** Inge Steegemans, Kassaye Sisay, Ernest Nshimiyimana.

**Formal analysis:** Inge Steegemans, Simon Collin, Emiliano Lucero, Koert Ritmeijer.

**Funding acquisition:** Turid Piening, Birhanu Sahelie, Dirk Essink, Koert Ritmeijer.

**Investigation:** Inge Steegemans, Kassaye Sisay, Ernest Nshimiyimana, Endale Menberu Tessema, Birhanu Sahelie.

**Methodology:** Inge Steegemans, Simon Collin, Emiliano Lucero, Koert Ritmeijer.

**Project administration:** Ernest Nshimiyimana, Gashew Gebrewold, Turid Piening, Endale Menberu Tessema, Birhanu Sahelie, Dirk Essink, Koert Ritmeijer.

**Supervision:** Ernest Nshimiyimana, Turid Piening, Dirk Essink, Emiliano Lucero, Koert Ritmeijer.

**Validation:** Inge Steegemans, Gashew Gebrewold, Gabriel Alcoba, Fikre Seife Gebretsadik, Dirk Essink, Simon Collin, Emiliano Lucero, Koert Ritmeijer.

**Writing – original draft:** Inge Steegemans, Emiliano Lucero, Koert Ritmeijer.

**Writing – review & editing:** Inge Steegemans, Kassaye Sisay, Ernest Nshimiyimana, Gashew Gebrewold, Turid Piening, Endale Menberu Tessema, Birhanu Sahelie, Gabriel Alcoba, Fikre Seife Gebretsadik, Dirk Essink, Simon Collin, Emiliano Lucero, Koert Ritmeijer.

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
