## [Decision Letter · Decision Letter 0]

27 Jul 2021

Dear Dr Steegemans,

Thank you very much for submitting your manuscript "Epidemiology, treatment outcomes and antivenom effectiveness in snakebite patients in north-west Ethiopia – a retrospective analysis." for consideration at PLOS Neglected Tropical Diseases. As with all papers reviewed by the journal, your manuscript was reviewed by members of the editorial board and by several independent reviewers. In light of the reviews (below this email), we would like to invite the resubmission of a significantly-revised version that takes into account the reviewers' comments. 

In particular, two reviewers have noted concerns about the statistical analysis/presentation of the results and multiple inconsistencies were noted in results tables. 

We cannot make any decision about publication until we have seen the revised manuscript and your response to the reviewers' comments. Your revised manuscript is also likely to be sent to reviewers for further evaluation.

Sincerely,

Stuart Robert Ainsworth

Associate Editor

Janaka de Silva

Deputy Editor

Reviewer's Responses to Questions

**Key Review Criteria Required for Acceptance?**

**Methods**

-Are the objectives of the study clearly articulated with a clear testable hypothesis stated?

-Is the study design appropriate to address the stated objectives?

-Is the population clearly described and appropriate for the hypothesis being tested?

-Is the sample size sufficient to ensure adequate power to address the hypothesis being tested?

-Were correct statistical analysis used to support conclusions?

-Are there concerns about ethical or regulatory requirements being met?

Reviewer #1: (No Response)

Reviewer #2: The objectives of the study were clearly stated. The study was appropriately designed to address the stated objectives and the study population was clearly described and appropriate to to test the hypothesis. Correct statistical analysis were used to arrived at the conclusion. All the ethical and regulatory requirement were met.

Reviewer #3: The objectives and study design are articulated and appropriate statistics used for analysis. Ethics approval in place.

That said the following needs attention:

1. The data collection tool needs to be provided - perhaps as a supplementary table

2. Patient inclusion/exclusion criteria need to be carefully described

3. the SAVP product - SAIMR poly is described in the introduction as being used by MSF in this hospital but is not included in the analysis - an explanation for this exlucsion is needed

4. A table describing the antivenoms used is needed (akin to that used by the Potet et al review of antivenoms) particularly with regards the venoms used for manufacture - as a supplementary table

**Results**

-Does the analysis presented match the analysis plan?

-Are the results clearly and completely presented?

-Are the figures (Tables, Images) of sufficient quality for clarity?

Reviewer #1: (No Response)

Reviewer #2: The analysis presented matched the analysis plan, and the results were clearly and completely presented. The tables and figures were of sufficient quality and clarity.

Reviewer #3: This is a very important paper that makes several important points about snakebite management in sub-Saharan Africa but it requires careful editing to maximise its impact. The grammar, spelling and phrasing needs attention:

1. the name of the ICP antivenom is EchiTAb-PLUS-ICP - correct this throughout.

2. antivenoms are often described as 'incorrect' or 'wrong' - please correct this inadequate description throughout and make it understandable to a non-snakebite audience - using a reference to the 'antivenoms' table suggested elsewhere in this review. 

I have the following questions regarding data management and results presentation: 

3. I'm not a statistician but isn't a probability of 0.000 meaningless? Should it be p<0.005?

4. much of the data/analysis is upon 'complications' but this term is inadequately defined - the methods section would benefit from a section defining this, and subsequent analyses.

5. Table 7 focuses upon 'complications' but includes only 1 distinct analysis criteria from those in Table 6, which focuses upon Rx outcomes - why is this duplication needed? 

6. And why are the figures for many of these identical criteria different in Tables 6 and 7?

7. In fact- please review both these Tables VERY carefully - because Table 6 (Rx outcome analysis) has a 'complications' analysis criteria but not a 'poor outcome' criteria, whereas Table 7 (Rx complications analysis) does NOT have a 'complications' analysis criteria. Please fix this major confusion and remove duplicated data (that are numerically inconsistent between the tables). 

8. The bar charts for Figure 1 should be in colour to enable greater clarity of the change in annual admission rates - this is an important part of the study's finding and deserves greater clarity

**Conclusions**

-Are the conclusions supported by the data presented?

-Are the limitations of analysis clearly described?

-Do the authors discuss how these data can be helpful to advance our understanding of the topic under study?

-Is public health relevance addressed?

Reviewer #1: (No Response)

Reviewer #2: The conclusions were supported by the data presented. The manuscript and the data presented are elaborate and help in advancing the understanding of the subject matter. The public health relevance has been addressed.

Reviewer #3: The conclusions are supported by the data analysis, with the exception that assigning snake species to hospital admissions (Table 1) is not supported by identification methods. If the authors want to include this (which I suggest they don't because it weakens the study's accuracy - see very speculative assumptions Abstract line 48/9), this entry should read 'Suspected snake species responsible for admission'). In this context - please be more careful using snake species names - the saw-scaled vipers in Egypt are either Echis coloratus or E. pyramidum - only the later is present in Ethiopia. It is correctly termed the North East African saw-scaled viper. 

Antivenom affordability is correctly described as a major issue in snakebite management - but is not analysed here. Can this data and analysis be included here, especially in terms of patient admission, outcome and adverse effects?

The higher rate of complications when using FavAfrique (line 349) deserves a much more comprehensive discussion - and was lower than Vacsera (Table 7).

Cite ref 35 for high adverse effects of the Vacsera product.

Paragraph starting line 304. The NE Nigeria snakebite-research team led by Prof Habib has also noted that admission rates increase when availability of antivenom becomes known in communities - I suggest you seriously consider citing ref 39 here (or other papers by this research team) to emphasise that this aspect of health seeking behaviour is evident in different settings. Its an important point to improve snakebite management. 

I have made several critical comments but only in an attempt to improve the impact and uptake of this very important paper that delivers many important points - that hopefully will receive attention and action from appropriate national, regional and international agencies.

**Editorial and Data Presentation Modifications?**

Reviewer #1: (No Response)

Reviewer #2: The authors looked at the effectiveness of 3 different antivenoms that were used at different periods on outcome of snakebite envenoming. Differences in time between admission and antivenom administration were reported across the three groups, however, it was not shown if this difference is statistically significant. This is relevant because time delay between hospital admission and antivenom administration negatively affect outcome. The authors need to address this and provide information on how the confounder effect was addressed.

Reviewer #3: Please review the grammar, spelling and phrasing throughout.

In addition to the many points already made:

Line 131 - SAIMR is manufactured with, and effective against, venoms from many Bitis vipers, including puff adders

I really dislike the inclusion of several non-authorative, non peer-reviewed, internet references in the list of citations. This seems lazy referencing to me - especially refs 10, 15. There exist several other, peer-reviewed papers that make the points that the authors seek to support their themes.

**Summary and General Comments**

Reviewer #1: Thank you for the opportunity to review “Epidemiology, treatment outcomes and antivenom effectiveness in snakebite patients in north-west Ethiopia – a retrospective analysis”. This is an important topic in an understudied area, and I think this paper will make a good addition to the literature. Although I think it’s quite interesting, I do have some problems with the statistics and feel revisiting this section with different tools would make these results more useful for clinicians and policy makers. 

Major comments:

Using Pearsons correlation is an unusual test to compare independent variables and your outcome of treatment (poor/ recovered). Pearsons works for normally distributed variables, and isn’t a great choice for binary outcomes; are you sure the variables meet these conditions? The correlation coefficient you present in Table 6 is hard to interpret and thus not very useful. Because you have a limited number of outcomes, you don’t have much statistical power for multivariable modeling, but I would recommend instead presenting the results from a logistic regression to show the associations between patient characteristics and outcome (i.e. one variable at a time). Presenting this as an odds ratio (or prevalence ratio, which is preferred though perhaps less common) with the 95% CI (and perhaps p value if you wish) will give the reader more informative data on the magnitude, precision, and significance of the association between each set of variables. You may wish to consider a limited multivariable analysis (e.g., one predictor variable, controlling for antivenom type and maybe time/year?) but you have a limited number of outcomes, so this may not be feasible. 

Regarding using ANOVA and your post-hoc tests, the post hoc tests use a selective component of the shared variance for all the groups taken from the overall F statistic. That said, the sample sizes in the groups are different, and Tukey-HSD test is for samples of the same size. You may wish to consult with your statistician, I think you probably should have used a Tukey-Kramer test. 

Line 237: Can you say anything about the significance of differences across groups of patients? It looks like the FAV-Afrique patients took longer to come in to the hospital. Is this a ‘statistically significant’ difference that you may wish to control for, or are these means and medians not meaningfully different? Since you have so few outcomes as to make multivariable analysis quite limited, I would urge you to compare your populations (i.e., show a p value across groups) so the reader can be aware of where differences lie. 

Table 3: I’d recommend showing the number and percent of persons with the outcome of interest, rather than the “mean” or proportion as is currently shown (i.e., first column shows total persons who received treatment, second column shows total number with positive outcome, third column shows the percentage XX.X% of persons with positive outcome). Same recommendation applies to Tables 4 and 5. 

Minor comments: 

When in 2015 did you begin providing antivenom (i.e., what month did you start administering Fav-Afrique)? It would be appreciated if you could make more precise your duration-of-use estimates in line 115-116. Could you also specify in this section of the Introduction: what was the standard of care prior to MSF’s arrival? 

Line 162: Please describe a bit more about those who did not receive antivenom. Did all in this category demonstrate symptoms (Table 1 suggests this is the case)? Is there anyone in this category who should have received antivenom but did not for whatever reason? What about patients who come who do not exhibit symptoms (e.g., they were bitten, but the envenomation was mild/nonexistent) – were there any patients who really didn’t need much help at all, but were just worried? Does this category also draw on patients admitted during the gap after Fav-Afrique expired and no treatment was available? I would assume this would then bias the type of patient included as a comparison (they’d be more likely to be in high need of antivenom even though they could not get it)

You mention the SAIMR polyvalent antivenom in line 127, but this is not shown as one of the antivenoms used in lines 115-116. Please clarify, or add this antivenom. (edit: I see you mention this in line 166. I would recommend that you clarify that too few patients were treated with SAIMR to be included in the analysis in lines 115-6)

Table 2: what is “defaulted”? Is that lost to follow up/unknown? Could you edit this last variable to match your outcome as defined in the methods (i.e., “cured” vs. “poor” – but including death as one of the “poor” outcomes as that’s informative)

Minor grammatical issues throughout, e.g., Line 36+: “Since 2015 Médecins Sans Frontières is treating” could be changed to “Since 2015 Médecins Sans Frontières has been treating”; Line 70 (and same comment for line 71): “This study shows that patients treated with VacSera antivenom have a higher chance on a poor outcome” should be “have a higher chance of a poor outcome” Lines 260 and 267 “Tuckey” should be “Tukey”. These are mostly minor, as I did not see any grammatical issues that interfered with my ability to interpret your message.

Reviewer #2: Data on the effectiveness and safety of the many antivenoms available in Africa is sparse and this has a huge implication on the management of snakebite envenoming in the continent. The manuscript therefore, provides a vital information on this subject matter. However, the retrospective nature of the study is a weakness as some vital information were not captured. Nevertheless, the authors must be recommended by providing an insight on this important issue and pave a way for further prospective study.

Reviewer #3: As stated before, I think this is an important paper to our filed, whose findings need to be widely disseminated and acted upon. 

I have made several critical comments but only in an attempt to improve the impact and uptake of this very important paper that delivers many important points - that hopefully will receive attention and action from appropriate national, regional and international agencies.

PLOS authors have the option to publish the peer review history of their article (what does this mean?). If published, this will include your full peer review and any attached files.

Reviewer #1: Yes: Matt A Price

Reviewer #2: No

Reviewer #3: No
---

## [Decision Letter · Decision Letter 1]

1 Nov 2021

Dear Dr Steegemans,

Thank you very much for resubmitting your manuscript "Epidemiology, treatment outcomes and antivenom effectiveness in snakebite patients in north-west Ethiopia – a retrospective analysis." for consideration at PLOS Neglected Tropical Diseases. As with all papers reviewed by the journal, your manuscript was reviewed by members of the editorial board and by several independent reviewers. In light of the reviews (below this email), we would like to invite a further re-submission of a significantly-revised version that takes into account reviewer #2's comments. 

We cannot make any decision about publication until we have seen the revised manuscript and your response to the reviewers' comments. Your revised manuscript is also likely to be sent to reviewers for further evaluation.

Sincerely,

Stuart Robert Ainsworth

Associate Editor

Janaka de Silva

Deputy Editor

Reviewer's Responses to Questions

**Key Review Criteria Required for Acceptance?**

**Methods**

-Are the objectives of the study clearly articulated with a clear testable hypothesis stated?

-Is the study design appropriate to address the stated objectives?

-Is the population clearly described and appropriate for the hypothesis being tested?

-Is the sample size sufficient to ensure adequate power to address the hypothesis being tested?

-Were correct statistical analysis used to support conclusions?

-Are there concerns about ethical or regulatory requirements being met?

Reviewer #1: - yes

- yes

- yes

- yes

- yes

- no concerns

Reviewer #4: Aims could be more clear: 

It is stated this study provides data on incidence - I don't think this is correct as there is no denominator data. The numbers presenting to hospital with snakebite is defined.

As this is a non-randomised study that aims to comapre safety and effectiveness of 3 antivenoms, much more needs to be done to adjust for baseline differences/confounders. The statistical analysis is basic and there is no adjustment for baseline differences between the groups.

There is insufficient information on how the outcomes were allocated as 'cured' vs 'poor outcome.' Was this a single person or two independent people? Was this person blinded to the treatment outcome? More information on how 'cured' was defined is needed: in particular, whether this was objective or open to interpretation.

**Results**

-Does the analysis presented match the analysis plan?

-Are the results clearly and completely presented?

-Are the figures (Tables, Images) of sufficient quality for clarity?

Reviewer #1: - yes

- yes

- yes

Reviewer #4: As highlighted above, major changes are required in the analysis. Table 1 should list all confounders that may account for differences in treatment outcome: particularly any differences in severity of envenoming at baseline. Degree of swelling, proprortion with systemic bleeding, and proportion with abnormal 20WBCT at baseline are important.

If there are differences betweent the groups at baseline then either comparisons of the antivenoms should not be done, or adjustment for the baseline differences between the groups needs to be undertaken. Consider matching, or more advanced statistical techniques for adjusting for baseline differences. As this is a non-randomised study, it is highly vulnerable to bias and this needs to be made up for with carefully considered analysis. I would suggest seeking further support from an experienced statistician. 

As this is retrospective, there must have been missing and innacurate data. This needs to be detailed.

I don’t think Spearman's Rho is the best analysis for Table 5. Multivariate logistic regression analysis would be preferrable. Reduce the number of predictor variables if there are issues with small numbers. Have missing data prevented undertaking multivariate logistic regression analysis?

**Conclusions**

-Are the conclusions supported by the data presented?

-Are the limitations of analysis clearly described?

-Do the authors discuss how these data can be helpful to advance our understanding of the topic under study?

-Is public health relevance addressed?

Reviewer #1: - yes

- yes

- yes

- yes

Reviewer #4: This may change when the analysis is updated.

**Editorial and Data Presentation Modifications?**

Reviewer #1: N/A

Reviewer #4: Suggest revising cited articles in first paragraph:

Reference 1 – avoid referencing opinion type articles for epidemiology. Consider Kasturiratne 2008.

Reference 3 seems incorrect for epidemiological statement?

**Summary and General Comments**

Reviewer #1: N/A

Reviewer #4: Overall this is an important and well written paper. The dataset is of a reasonable sample size and seems to be of good quality. Unfortunately I think the study cannot be published without making fairly major changes to the analysis. Non-randomsied studies such as this are highly vulnerable to bias and careful analysis is needed to overcome this. This could change the main finding of the paper. Comparing crude proportions with 'cure' vs 'poor outcome' can be misleading. Once this is done, the paper would be of high quality and would certainly be suitable for publications in PLOS NTD.

PLOS authors have the option to publish the peer review history of their article (what does this mean?). If published, this will include your full peer review and any attached files.

Reviewer #1: Yes: Matt Price

Reviewer #4: Yes: Michael Abouyannis
---

## [Decision Letter · Decision Letter 2]

5 Jan 2022

Dear Dr Steegemans,

We are pleased to inform you that your manuscript 'Treatment outcomes among snakebite patients in north-west Ethiopia – a retrospective analysis.' has been provisionally accepted for publication in PLOS Neglected Tropical Diseases.

Best regards,

Stuart Robert Ainsworth

Associate Editor

Janaka de Silva

Deputy Editor

Reviewer's Responses to Questions

**Key Review Criteria Required for Acceptance?**

**Methods**

-Are the objectives of the study clearly articulated with a clear testable hypothesis stated?

-Is the study design appropriate to address the stated objectives?

-Is the population clearly described and appropriate for the hypothesis being tested?

-Is the sample size sufficient to ensure adequate power to address the hypothesis being tested?

-Were correct statistical analysis used to support conclusions?

-Are there concerns about ethical or regulatory requirements being met?

Reviewer #4: Yes

**Results**

-Does the analysis presented match the analysis plan?

-Are the results clearly and completely presented?

-Are the figures (Tables, Images) of sufficient quality for clarity?

Reviewer #4: Yes

**Conclusions**

-Are the conclusions supported by the data presented?

-Are the limitations of analysis clearly described?

-Do the authors discuss how these data can be helpful to advance our understanding of the topic under study?

-Is public health relevance addressed?

Reviewer #4: Yes

**Editorial and Data Presentation Modifications?**

Reviewer #4: Nil

**Summary and General Comments**

Reviewer #4: This is a well written and important study, which provides key insight into the burden of snakebite in this region of Ethiopia. I commend the authors for updating the analysis, which is greatly improved. The comparisons between the antivenom products are fair and valid, with the inherent limitations of non-randomized data clearly defined in the discussion.

My only minor reccomendation would be to include the full results of the univariate and multivariate logistic regression analyses that are summarized in Tables 3 and 4. The unadjusted +/- adjusted ORs and CIs for each covariate should be listed, as many readers will be interested in this. This can be as a supplementary table in the appendices.

With this minor addition, I would suggest this paper is accepted for publication.

PLOS authors have the option to publish the peer review history of their article (what does this mean?). If published, this will include your full peer review and any attached files.

Reviewer #4: **Yes: **Michael Abouyannis

---

## [Editor Report · Acceptance letter]

4 Feb 2022

Dear mrs Steegemans,

We are delighted to inform you that your manuscript, "Treatment outcomes among snakebite patients in north-west Ethiopia – a retrospective analysis," has been formally accepted for publication in PLOS Neglected Tropical Diseases.

Best regards,

Shaden Kamhawi

co-Editor-in-Chief

Paul Brindley

co-Editor-in-Chief
